# Voice Quality Evaluation in a Mobile Cellular Network: In Situ Mean Opinion Score Measurements

**DOI:** 10.3390/s24206630

**Published:** 2024-10-15

**Authors:** Sorin Leonte, Andra Pastrav, Ciprian Zamfirescu, Emanuel Puschita

**Affiliations:** 1Communications Department, Technical University of Cluj-Napoca, 28 Memorandumului Street, 400114 Cluj-Napoca, Romania; sorin_leonte31@yahoo.com (S.L.); andra.pastrav@com.utcluj.ro (A.P.); 2Department of Telecommunications, Politehnica University of Bucharest, 1-3, Iuliu Maniu Ave., 061071 Bucharest, Romania; ciprian.zamfirescu@upb.ro; 3National Institute for Research and Development of Isotopic and Molecular Technologies, 67-103 Donat Street, 400293 Cluj-Napoca, Romania

**Keywords:** circuit switch (CS), mobile cellular network, mean opinion score (MOS), OTT application, packet domain (PS), voice over LTE (VoLTE), voice services

## Abstract

This article aims to test, measure and evaluate the quality of voice calls made in a mobile cellular network. A set of drive tests were conducted, during which logs were collected using specialized measurement terminals equipped with a dedicated voice evaluation application. Three different scenarios were considered: the first scenario consisted of a series of mobile-to-mobile calls in a circuit-switched (CS) domain over the GSM network, the second scenario involved similar calls using the VoLTE service in a packet-switched (PS) domain of a 4G network, and the third scenario employed an over-the-top (OTT) media service type via the WhatsApp application in the same PS domain of the 4G network. The measurement results highlight the user experience in each scenario and compare the voice quality evaluated through the Mean Opinion Score (MOS) across the CS and PS domains. The originality of this work consists of in situ measurements performed in Bucharest, Romania, providing detailed, context-specific insights regarding the network performance that can drive local improvements and support policy and investment decisions.

## 1. Introduction

Today, mobile telephony refers specifically to the ability to be consistently connected to a highly secure network that facilitates extensive data transfers and high-quality voice communications. A mobile cellular network’s performance can be assessed through both Key Performance Indicators (KPIs), reported by mobile devices to the network operator, and drive tests [1].

When assessing the voice quality, the user expects a very good audio experience that accurately reproduces the voice of the person on the other end. However, voice quality is a complex phenomenon within the process of human perception and is always subjective [2]. In this context, the speech quality is defined as a measure of the listener’s satisfaction, based on their experience and expectations regarding voice communication. It is generally expressed as a Mean Opinion Score (MOS), being a quality of experience (QoE) metric. This parameter indicates the average of many individual opinions on voice quality, obtained from a representative number of listeners who have assigned audio quality scores to a large set of voice samples.

The International Telecommunication Union—Telecommunications (ITU-T) Standardization Sector defined the ITU-T P.863 Perceptual Objective Listening Quality Assessment (POLQA) [3], a sophisticated method for objectively assessing the speech wave signals in a telecommunication context. The ITU-T P.863 POLQA can be considered the most successful and precise representative of speech quality prediction algorithms validated in the course of International Standardization [2].

POLQA accurately models hearing and perception and can be used with any speech stimulus of any language, being transparent to transcoding, noise floors or repackaging [4]. Therefore, it is suitable for evaluating the voice quality in various mobile network technologies, domains and applications. Such an approach is truly advantageous since different mobile technologies provide different voice services.

Second-generation networks support voice calls only in the circuit-switched (CS) domain. Third-generation networks use both CS and packet-switched (PS) domains for voice applications, while 4G and 5G networks provide voice over LTE (VoLTE)/voice over new radio (VoNR) services in the PS domain. Moreover, given the nationwide 4G coverage in many countries and the high mobility of individuals traveling abroad, over-the-top (OTT) applications have seen significant development, leading to a growing preference for these applications for voice services [5]. These applications employ voice over internet protocol (VoIP) technology to enable voice calls.

Within the mobile network, the user equipment (UE) typically prefers the most advanced technology provided by the operator.

In Europe, 5G technology is under deployment (e.g., in Romania, 5G coverage encompasses 32.5% of populated areas [6]). So, 4G is the go-to technology for simultaneous high-throughput data transfers and high-quality voice calls. However, in scenarios of poor coverage or significant packet loss, call continuity must be ensured through the Single Radio Voice Call Continuity (SRVCC) procedure [7]. As the increasing demand for a spectrum for 4G technologies has led to the decommissioning of 3G technologies [8], 2G (i.e., Global Service for Mobile Communications, GSM) is the CS domain technology ensuring call continuity under conditions of degraded signal quality.

In this context, the scope of our work is to evaluate through drive tests the voice service quality in 2G and 4G cellular mobile networks. POLQA is used to assess the quality of sequences of short mobile-to-mobile calls (i.e., of 80 s) in three different scenarios: (1) over the CS domain using 2G technology, and over the PS domain using (2) VoLTE and (3) the OTT WhatsApp version 2.23.12.75 application using 4G technology. Each drive test was performed with a car on a route of 10 km, conducting MOS measurements for approximately 25 min. Two UEs running the QualiPoc application are used as devices under test (DUTs).

The originality of this work consists of in situ measurements performed in Bucharest, Romania, providing detailed, context-specific insights regarding the network performance that can drive local improvements and support policy and investment decisions. Despite the growing body of research on voice quality evaluation in mobile networks, many studies are limited to controlled environments or focus on single technologies. This leaves a gap in the understanding of how voice services perform under real-world conditions, particularly when comparing legacy technologies like 2G with more advanced systems like VoLTE and OTT applications such as WhatsApp. This study addresses this gap by providing comprehensive, in situ measurements using a professional testing system widely employed by mobile operators. The real-world data collected through drive tests across various scenarios offer insights into the performance of different voice services under diverse conditions, contributing new knowledge that can help improve network optimization and user experience.

The rest of this paper is organized as follows. Section 2 presents the state of the art regarding speech quality assessment. Section 3 introduces the ITU-T P.863 POLQA standard for voice quality evaluation. Section 4 describes the MOS metric in voice quality evaluation. Section 5 presents the experimental setup and test configuration, while Section 6 indicates the measurement results. Finally, Section 7 concludes the paper.

## 2. The State of the Art

The methods used to assess speech quality in communications networks can be divided into three categories: subjective, objective intrusive, and objective non-intrusive. The first type involves testing with real users that assign MOSs to voice transmissions to evaluate the quality of their experience. This method is the most accurate one. The second type compares an original voice sample (reference voice sample) to a transferred sample that was transmitted in a communications network. This method is based on algorithms that use psychoacoustic models of human perception and provides results comparable to the subjective method. Finally, the third type does not need to use a reference signal. In this case, an MOS is calculated only from the parameters of the transferred sample. However, this method, while easy to implement, provides less accurate results.

The specialized literature presents many studies that evaluate the speech quality in mobile telephony networks, using various methods, for both circuit-switched and packet-switched voice services.

In [4], subjective measurements were performed in a GSM network, assessing the ‘Speech Intelligibility’ and the reduction in voice quality. The evaluation metrics used were the Percent Correct Words value and the Degradation Mean Opinion Score (DMOS).

In [9], the authors presented an objective MOS evaluation based on POLQA for Adaptive Multi-Rate Wideband (AMR-WB) Uplink in VoLTE calls, under controlled conditions. The test results showed that POLQA had a high sensitivity to the audio level and environment.

The work in [10] aimed to evaluate the performance of the ITU-T P.863 POLQA in the case of low-bitrate coders (such as Mixed-Excitation Linear Prediction, MELP) when using the Chinese language, and to compare its performance to its predecessor, the ITU-T P.862 Perception Evaluation of Speech Quality (PESQ) [11]. The work presented the results for both the subjective (i.e., with human users) and objective (i.e., using POLQA and PESQ) tests, but in a controlled environment, not in real network traffic conditions.

Regarding OTT applications, the work in [12] evaluates the performance of WhatsApp and Skype in terms of data consumption and voice quality using the PESQ testing standard. The tests were performed in various network conditions, increasing the packet loss, showing that both applications have a similar MOS in good network conditions, while WhatsApp performs better than Skype if the packet loss is greater than 20%.

Many attempts are made to create more reliable non-intrusive methods for speech quality assessment. The work in [13] proposes a non-intrusive estimator based on recurrent neural network architectures to predict the POLQA score of a speech signal in a short time context. Ref. [14] presents the design of a wireless analytical tool that models mobile voice quality by crowdsourcing and mining the network indicators of cellphones. The data are collected via user crowdsourcing through a client app, the model is calibrated according to the POLQA standard, and the performance of the tool is tested in a VoLTE network. Another approach is presented in [15], where a packet-level measurement method was introduced to measure the network and service quality online and provide an estimation model of a derivative MOS index. The proposed method does not inspect the voice payload and does not need the reference voice samples. The performance is validated against a full reference voice quality estimation model called AQuA, using real VoIP traffic in controlled network scenarios. Ref. [16] aims to provide an accurate method for assessing speech quality by predicting the MOS using machine learning techniques, and specifically convolutional neural networks. This approach is primarily algorithmic and predictive, relying on predefined audio samples. The work in [17] focuses on building QoE models based on delay and jitter for 4G and 5G networks, exploring how network conditions affect user experience, particularly in terms of service degradation. The study employed subjective methods like surveys to construct QoE models. Ref. [18] used artificial neural networks (ANNs) for predicting the voice quality in VoLTE and VoNR applications. The performance of the proposed model was evaluated against MOS values obtained in several drive tests, emphasizing a data-driven, predictive approach rather than real-world measurements. The research in [19] evaluates the 5G network’s capability to support voice over new radio (VoNR) using intelligent MOS predictions based on deep neural networks. The proposed scheme is validated against drive test data.

The work in [20] compared a new objective speech quality model, Virtual Speech Quality Objective Listener (ViSQOL), to the PESQ and POLQA standards. The study relied on controlled laboratory simulations using predefined datasets (NOIZEUS and E4), where the focus is on comparing the predictive accuracy of speech quality models in a range of synthetic conditions.

A more comprehensive study on VoLTE performance evaluation is presented in [21]. The VoLTE performance in commercial networks was evaluated in terms of the Robust Header Compression (RoHC), Transmission Time Interval (TTI) bundling, Real-Time Transport Protocol (RTP) error rate, jitter, Block Error Rate (BLER), handover delays (C-plane and U-plane), and voice quality in terms of the MOS. The study compares the VoLTE performance against 2G and 3G voice transmissions and OTT applications. The POLQA standard is used in the field tests and it shows that VoLTE ensures the best performance.

The work in [22] focuses on using the MOS to assess the performance of video transmissions. It proposes a Bayesian network model to predict the MOS for video, using factors like the bit rate and latency.

Ref. [23] studies how roaming across different European networks impacts users’ quality of experience, focusing on broader QoE issues across multiple regions.

Our paper proposes an experimental approach based on drive test measurements to assess the performance and compare the voice MOSs for calls made using GSM, VoLTE, and WhatsApp.

### Key Strengths of Our Work

Compared to the studies presented above, our work has many strengths, as follows.

First, our work includes real-world measurements collected through drive tests. Unlike many other studies [13,14,15,16,17,18,19,20,22] that rely heavily on simulations, predictive models or laboratory tests, our work offers a more comprehensive, real-world perspective on the network performance in diverse conditions, as it uses drive tests over a 10 km route to provide valuable insights into how the network performance fluctuates in real environments, which is crucial for operators seeking to improve service reliability.

Second, our study offers a broader comparison by evaluating three distinct voice technologies: GSM (CS), VoLTE (PS) and OTT services. This enables us to provide insights into how different network architectures affect user experience. This stands in contrast to other articles [4,9,10,12,13,14,15,16,17,18,19,20], which often focus on only one technology or service, limiting the scope of their analysis.

Third, our work uses an objective approach, using standardized POLQA-based technical measurements made during drive tests. As such, it provides practical data that operators can directly use to improve the network performance, rather than relying solely on subjective user feedback [4] that is cumbersome and cannot be used by operators to assess the network conditions and performance in a timely manner.

Fourth, our work focuses on the voice quality in a localized environment, providing detailed assessments of the GSM, VoLTE and OTT performance in Bucharest, unlike other studies [23] that focus on broader QoE issues across multiple regions. The in situ analysis of the voice quality in our article provides network operators with actionable insights. These insights help identify performance gaps and enable improvements to user experience based on the actual network conditions.

## 3. Speech Quality Measurements with POLQA ITU-T P.863

The ITU-T defined ITU-T P.863, POLQA, as a sophisticated method for objectively assessing the listening quality of speech signals, typically used in telecommunications to evaluate the performance of various network components. The current ITU-T standard P.863, POLQA, is considered the most successful and precise representative of speech quality prediction algorithms [4].

### Transmision Path and Modeling in POLQA ITU-T P.863

Today, ITU-T P.863 stands as the benchmark method for assessing voice quality. It has been deployed across numerous measuring instruments worldwide for the purpose of quality monitoring. The standard is continuously updated through regular model maintenance. Published in March 2018, version 3.0 underwent specialized verification to accommodate the new Enhanced Voice Services (EVS) coding method. It now encompasses the entire audible spectrum for humans. It is recommended to use ITU-T P.863 “POLQA v3” in the full-band mode for all types of voice quality measurements in telecommunications and mobile networks [4].

The ITU-T P.863 approach is called full-reference, which means that the quality prediction is based on the comparison between an undistorted reference (original) signal and the received signal. To evaluate mobile communication channels, a reference speech signal from a preconfigured remote station is received, recorded, and compared to a local copy of the reference signal; this constitutes a typical end-to-end measurement.

The simplified structure of ITU-T P.863 is illustrated in Figure 1.

The measuring systems are primarily designed for mobile use, aimed at assessing the quality of voice connections in real networks while in motion. Simplified, ITU-T P.863 encompasses three aspects: the preprocessing and synchronization of the reference and test signals, the modeling of auditory physiology, and the modeling of speech perception and temporal integration.

All the analyses and quality value calculations are based solely on the speech signal itself. ITU-T P.863 does not need any additional information, such as IP data, and treats the transmission channel as a black box. Signal preprocessing covers basic technical steps like correcting clock frequency deviations, adjusting signal levels and prefiltering as needed to model the auditory environment. The most critical and complex aspect of speech quality measurement is precisely synchronizing the reference and test signals for analysis in short time windows. Once aligned, the signals are mapped in overlapping windows (Fast Fourier Transform, FFT) and psychoacoustically transformed to create a time–frequency representation for hearing. This is followed by converting sound events into internal stimuli, forming the core of the ITU-T P.863 speech-processing model.

Certain hearing phenomena, such as the absolute threshold of hearing, are well known. This threshold, which is frequency-dependent, represents the point below which sounds cannot be heard. However, hearing physiology is more complex and involves effects like spectral and temporal masking.

The goal of psychoacoustic algorithms like ITU-T P.863 is to generate a signal similar to what the auditory nerve delivers to the brain’s speech-processing center. Instead of calculating the difference between the original and distorted transmitted signal at the amplitude level, ITU-T P.863 focuses on differences in the “internal representation” of the signals—what is perceived by the brain after auditory processing. This approach filters out inaudible components. While it might seem that simply accumulating noticeable differences in frequency and time would provide a quality assessment, this is not enough. Postprocessing these differences is needed, relying on the brain’s ability to not just understand speech but also perceive its naturalness. Finally, the calculated quality value is mapped to the MOS scale from 1 to 5 [2].

## 4. MOS Parameters in Voice Quality Evaluation

The network architecture of GSM relies on CS, whereas LTE uses a PS-based architecture. The fundamental distinction between CS and PS lies in how data transmission is handled: in the CS mode, a physical channel is allocated until data transmission commences, establishing a dedicated communication path between the sender and receiver. In contrast, PS eliminates the necessity for transmitting messages via a dedicated path. The data in PS are segmented into packets and then grouped together, with each packet individually routed from the source to the destination. Therefore, due to LTE’s PS-based architecture, the IP Multimedia Subsystem (IMS) was introduced as an LTE network resource, facilitating the transmission of voice calls using the IP protocol [9]. Within the context of an LTE mobile network, an OTT voice app uses the mobile device’s data connection to the internet over the LTE network to offer a voice service by a third party that is independent of the mobile service provider. An example of such OTT voice apps is WhatsApp [24].

The speech quality can be degraded by the way the voice is encoded and transmitted, transmission delays, bandwidth limitations, packet losses, etc. For this reason, the voice quality is a complex measure that quantifies all influences on the voice quality on an absolute scale, resulting in a single number between 1 (bad) and 5 (excellent) with an accuracy of two decimal points. Note that an MOS of zero does not exist, and there is no MOS < 1. At the upper end, the scale ends at 5. Due to psychological reasons underlying listening experiments, in practice, the best MOS for a perfect, undistorted speech signal is slightly below 5, reaching 4.8. The listening quality is also reported as the LQ category or quality.

Table 1 describes the correspondence between the MOS values and LQ categories [25].

## 5. Experimental Setup

### 5.1. QualiPoc Test Application

The tests were conducted using a measurement system acquired from Rohde & Schwarz (Munich, Germany), which is a partner in the development of the POLQA standard. The system includes two DUTs running the Android operating system. These DUTs run QualiPoc, a measurement and evaluation application for voice and data mobile networks that implements the POLQA (W) standard [25].

Figure 2 shows the UEs used as DUTs in the drive tests. The picture shows the configuration screen (left) and the call window during operation (right).

The “Double-Ended Call” test was used to establish a voice connection and evaluate the speech quality between the two DUTs. In double-ended tests, both sides (i.e., A and B) are controlled by the measurement system and possess identical functionality. This means that either side can originate and terminate calls and can send and receive speech samples. The reference file used in the measurement tests was “En_P501_FM_wide_ref.waw”.

Table 2 indicates the call window configuration for the “Double-Ended Call”.

### 5.2. Drive Test Route and Scenarios

The measurements were conducted during a set of driving tests with a car. The DUTs were located inside the car on the dashboard. The chosen route for evaluating the speech quality and the corresponding MOS has a length of approximately 10 km, and the duration of one pass was approximately 25 min.

Figure 3 shows the drive test route.

The test measurements were performed in three different scenarios. The first scenario aimed to evaluate the performance of mobile-to-mobile calls in the CS domain over the GSM network. The second scenario envisaged similar calls using the VoLTE service in a PS domain of a 4G network. The third scenario employed an OTT media service type via the WhatsApp application in the same PS domain of the 4G network. The same call window and the same reference files were used for all three test scenarios.

For the VoLTE and WhatsApp drive tests, the DUTs were camped in LTE with the possibility to add new radio (NR) frequencies in the 5G Non-Standalone (NSA) scenario. Since the VoIP packet size, both in VoLTE and WhatsApp, is small, only the 4G network was used for the voice drive test. The LTE voice packets were routed only in the primary 4G cell, without the need to add NR frequencies. During the VoLTE and OTT drive tests, we identified a significant number of Physical Cell Identities (PCIs), indicating many 4G cells in the area chosen for the drive test. Additionally, we observed that the measurements indicated four distinct frequency bands: B3 [20 MHz], B3 [10 MHz], B1 [10 MHz], and B20 [10 MHz]. For the calls made in the GSM, the DUTs using 2G technology were blocked. The logs for the three scenarios were extracted using the Rohde & Schwarz Smart Analytics system. After the logs were extracted, the data were processed into graphs or displayed on a map.

## 6. Drive Test Results and Performance Evaluation

### 6.1. MOS Evaluation

During the tests, an MOS sample was evaluated once every 9 s. The three drive tests resulted in 86 MOS samples collected in the GSM, 95 MOS samples for the VoLTE service and 89 MOS voice samples for the OTT service over WhatsApp. The variable number of MOS samples is due to the variable duration of the three drive tests.

Figure 4 illustrates the average MOS in the three scenarios.

It can be observed that the highest MOS was obtained for VoLTE; most of the samples were classified as “Excellent”. This result is not surprising because the IMS server provides good QoS support for voice services in LTE. VoLTE in cellular mobile networks is a guaranteed service (Guaranteed Bit Rate, GBR) that operates on a different, prioritized QoS class indicator (QCI) compared to with usual data and OTT services. Additionally, the existence of a different QCI also allows for differentiation in mobility strategy compared to other types of services.

Mobile phone operators hold licenses to operate in multiple frequency bands for 4G technologies. In our drive tests, we identified 4 frequencies (layers) used for 4G cells. These layers (frequencies) are selected by the mobile device based on the mobility strategy implemented in the network, which typically relies on the signal levels reported by the mobile devices.

A VoLTE user typically benefits from a different mobility strategy compared to regular data users because a faster transition between 4G layers is desired, along with the avoidance of areas with poor signal and quality. Moreover, the VoLTE service benefits from new features such as TTI bundling and RoHC.

TTI bundling relies on sending multiple copies of the same packet within a short period (typically within one TTI, which is 1 ms in LTE), thus increasing the likelihood that at least one packet is successfully received and improving the reliability. TTI bundling helps reduce latency by minimizing the need for retransmissions, which is critical for real-time services like VoLTE. The RoHC feature is designed to maintain effectiveness even in environments where there is high packet loss or variability, ensuring robust performance. This means that RoHC can maintain compression and decompression processes reliably, even when network conditions are challenging [26].

Figure 5 shows the MOS values (top) and the Signal-to-Interference-plus-Noise Ratio (SINR) values in dB (bottom) along the VoLTE drive test route. It can be observed that despite the SINR samples being below 5 dB in some areas, most of the MOS values are still classified as excellent. This demonstrates the effectiveness of the RoHC and TTI bundling features.

For the OTT scenario, 15.7% of the samples were categorized as fair and 20.5% as good because of some variable delay and packet loss. Figure 6 shows the MOS (top) and SINR (bottom) values for the WhatsApp drive tests. It can be observed that sometimes, in areas with a lower SINR, MOS values between 3 and 4 occur. This degradation of the MOS values compared to the VoLTE results can be explained by the absence of TTI bundling and RoHC, correlated with the fact that WhatsApp packets are classified in the network like any other data packets.

Table 3 presents the measurements results divided into MOS categories.

In Table 3, For GSM, there are three samples classified as “Bad”, 17 as “Poor”, 15 as “Fair”, and 51 as “Good”. This indicates that while most of the GSM samples are of good quality, there are a notable number of lower-quality samples as well. It should be noted that the voice traffic in GSM networks is low, most often being used only when SRVCC is triggered due to the signal level or packet loss. The logs also show that the calls were made in full rate using the latest codec, AMR-WB, with a transfer rate of 12.65 kbps.

In the VoLTE scenario, 1 sample is categorized as “Fair”, 1 as “Good” and 93 as “Excellent”. This shows that VoLTE delivers a very high quality, with the vast majority of the samples rated as excellent and only two falling below that standard.

The majority of the WhatsApp calls were rated as “Excellent” (57 samples), with 18 samples rated as “Good” and 14 samples rated as “Fair”, indicating moderate quality, but there are no samples classified as “Bad” or “Poor”.

Figure 7 illustrates the distribution of Mean Opinion Score (MOS) samples for GSM calls across the quality categories: 3.5% of the samples fall into the “Bad” category (1.0–1.99), indicating very poor voice quality; 20% of the samples are classified as “Poor” (2.0–2.99), showing that a significant portion of calls experienced subpar quality; and 17.5% of the samples are rated as “Fair” (3.0–3.99), reflecting moderate voice quality.

The majority of the samples, 59%, are in the “Good” category (4.0–4.49), signifying the generally acceptable quality of the GSM calls.

There are no samples rated as “Excellent” (4.5–5.0), suggesting that the GSM calls did not reach the highest standard of quality in this scenario. Overall, while most of the calls achieved good quality, around 41% of the samples indicated some level of quality degradation. However, the poor and bad samples were not successive but interspersed with good samples, which diminishes the poor user experience.

Table 4 outlines the causes of quality degradation across different MOS categories for GSM calls, highlighting the specific technical issues affecting voice quality. For the “Poor” category calls, the main cause of degradation was interruptions, with 13 instances recorded. These interruptions, likely caused by handover issues or frame errors, significantly impacted the user experience. Additionally, the frame/packet loss and variable delay each contributed to one instance of poor quality, while two cases were categorized under “Not to Specify”, meaning no dominant cause was identified for the quality issues. In the “Bad” category, both interruptions and frame/packet loss were the key factors, each causing a few instances of degradation. These issues severely affected the voice quality, leading to a small number of badly rated calls.

For the calls rated as “Fair”, background noise and interruptions were the primary causes, with each contributing to one instance of degraded quality. In this category, four instances were marked as “Not to Specify”, indicating more complex or multiple factors affecting the quality. Additionally, nine instances of fair-quality calls were rated as “OK”, meaning that while the calls were not perfect, no specific issues were noted. In the “Good” quality category, the majority of calls (51 instances) were rated as “OK”, with no significant problems affecting the call quality. This suggests that most of the GSM calls in this group were satisfactory, with users experiencing no major disruptions. Overall, interruptions and packet loss emerge as the primary causes of degradation, particularly in the “Poor” and “Bad” categories. However, the “Good” calls show that a significant number of the GSM calls maintained a high level of performance, with minimal issues reported. The presence of “Not to Specify” in the “Fair” and “Poor” categories indicates that in some cases, multiple factors or less identifiable causes contributed to the degradation of the call quality.

Figure 8 clearly shows that VoLTE offers an outstanding performance, with almost all of the samples (98%) rated as “Excellent”. Only a very small percentage of calls (1% each) fell into the “Fair” and “Good” categories, and no calls were rated as “Bad” or “Poor”. This demonstrates VoLTE’s ability to provide consistently high-quality voice services. All the samples are rated as “OK”, and even the “Fair” and “Good” samples do not present any significant issues.

Figure 9 shows that 64% of the WhatsApp samples were rated as “Excellent”, 20% as “Good”, and 16% as “Fair”, indicating that the majority of the calls were of a high quality, with no samples in the “Bad” or “Poor” categories.

Table 5 shows the quality causes for the WhatsApp calls across the quality categories. For the “Fair” category, nine cases were due to frame/packet loss, where the loss of data packets likely affected the call quality; three cases were caused by variable delay, indicating inconsistent timing in data transmission, two cases had no significant issues and were rated as “OK”. In the “Good” category, 17 samples were classified as OK, meaning no noticeable issues were found, and in the “Excellent” category, 57 samples were rated as “OK”, representing high-quality calls without any significant degradation. The table highlights that frame/packet loss and variable delay are the primary causes of degradation in the calls rated as “Fair”. In contrast, the majority of the calls rated as “Good” or “Excellent” did not encounter noticeable issues, indicating a high level of performance for most of the calls.

### 6.2. Performance Evaluation

The results demonstrate that VoLTE consistently delivers high-quality voice services, with most samples categorized as “Excellent”. This is largely due to the dedicated QoS mechanisms that VoLTE benefits from, such as GBR and specific QoS class indicators, which prioritize voice traffic over other data. VoLTE also employs advanced features like TTI bundling and RoHC, which further enhance the voice reliability by reducing packet loss and improving latency in challenging conditions. These features allow VoLTE to maintain an excellent performance even in areas with a suboptimal SINR.

Given these features, VoLTE emerges as the most reliable option for voice communication in modern mobile networks, particularly in scenarios where high voice quality is critical, such as emergency services. Additionally, VoLTE’s ability to transition seamlessly through the SRVCC procedure when 4G coverage is lost ensures that voice calls are maintained, providing an added layer of reliability.

The WhatsApp service, although not as technically robust as VoLTE, still performs well in most cases. A significant portion of the WhatsApp call samples were rated “Excellent” and “Good”, showing that OTT services can provide an acceptable voice quality when network conditions are favorable. However, WhatsApp’s performance is more variable than that of VoLTE, with a notable number of samples rated as “Fair”, indicating moderate quality degradation.

The absence of advanced features like TTI bundling and RoHC in WhatsApp calls makes the service more vulnerable to packet loss and variable delay, which can affect the user experience, particularly in areas with a lower SINR. Additionally, as an OTT service, WhatsApp relies entirely on the quality of the underlying data connection and does not benefit from the same prioritization mechanisms as VoLTE. Consequently, WhatsApp calls are more susceptible to call drops when the data connection is unstable or lost, as there is no SRVCC procedure to ensure call continuity.

The GSM network continues to play an important role, particularly in ensuring call continuity through SRVCC when 4G coverage is insufficient. The majority of the GSM call samples were rated as “Good”, demonstrating that GSM can still provide satisfactory voice quality, particularly when modern codecs like AMR-WB are used. However, the variability in the GSM performance is notable, with a significant number of samples classified as “Fair” and some even rated as “Poor” or “Bad”.

The degradation in the GSM call quality can be attributed to interruptions during handovers, frame/packet loss, and background noise, which are more prevalent in CS technologies. Despite these limitations, GSM remains a crucial fallback technology, especially in rural or poorly covered areas, where modern packet-switched services like VoLTE may not be available.

When comparing the three technologies, it is evident that VoLTE offers the highest and most consistent voice quality, primarily due to its guaranteed QoS mechanisms and advanced features designed specifically for voice services. WhatsApp, while versatile and widely accessible, cannot match VoLTE in terms of reliability and consistency, particularly in challenging network environments. However, it remains a valuable alternative for users who prioritize cost-effective communication over guaranteed quality.

GSM, on the other hand, plays a supporting role, ensuring that voice services are maintained in areas with poor 4G coverage. Although it does not deliver the same high-quality experience as VoLTE or even WhatsApp in many cases, its ability to provide continuity through SRVCC makes it a vital component of mobile voice infrastructure.

## 7. Conclusions

This study aimed to assess the voice quality in mobile cellular networks using in situ MOS measurements across three different technologies: GSM, VoLTE and OTT services. The evaluation provides significant insights into how each technology performs under real-world conditions, contributing to a broader understanding of the strengths and limitations of each service.

The findings of this study highlight the clear advantages of VoLTE as the premier voice communication technology in modern mobile networks, thanks to its superior QoS mechanisms and advanced voice optimization features. WhatsApp serves as a valuable secondary option, especially for users seeking free or low-cost communication services, though it remains more vulnerable to network variability. GSM, while no longer the primary choice for voice communication, continues to ensure call continuity and coverage in more challenging network environments.

The results of this study provide critical insights for network operators and policy makers, emphasizing the need to further invest in VoLTE and similar technologies that guarantee high-quality voice communication. At the same time, it underscores the importance of maintaining GSM as a reliable fallback option, particularly in areas where 4G or 5G infrastructure is not fully developed. As mobile voice services continue to evolve, ensuring consistent and high-quality voice communication will remain a key focus, both for user satisfaction and for maintaining the overall reliability of telecommunications infrastructure.

## Figures and Tables

**Figure 1 sensors-24-06630-f001:**
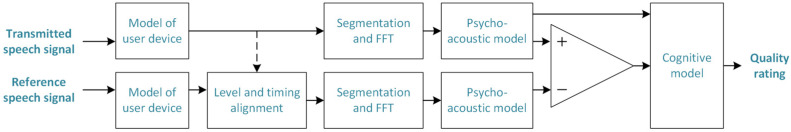
Simplified structure of ITU-T P.863.

**Figure 2 sensors-24-06630-f002:**
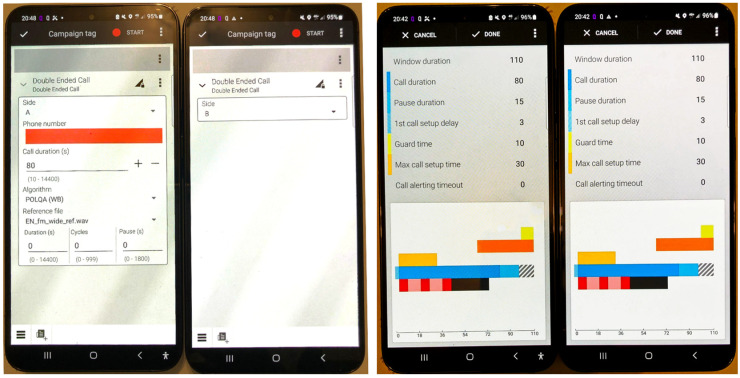
A-side and B-side DUTs used for tests: call configuration (**left**) and call window (**right**).

**Figure 3 sensors-24-06630-f003:**
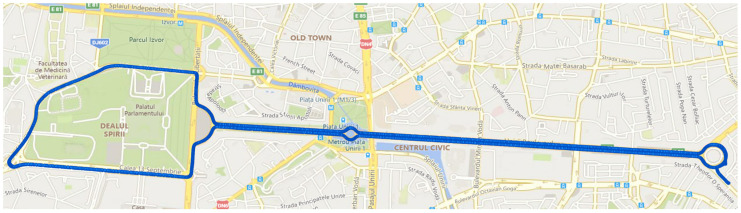
Drive test route in Bucharest, Romania.

**Figure 4 sensors-24-06630-f004:**
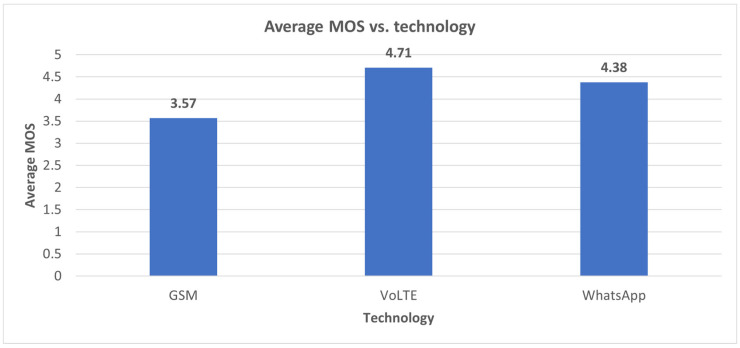
Average MOS measured.

**Figure 5 sensors-24-06630-f005:**
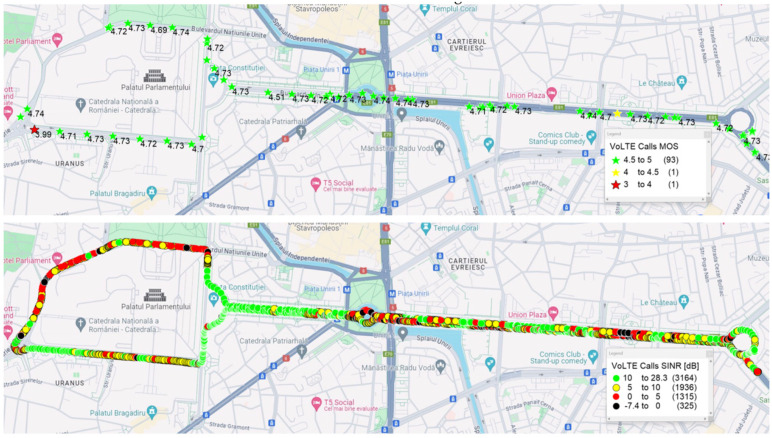
VoLTE drive test: correlation between MOS (**top**) and SINR (**bottom**).

**Figure 6 sensors-24-06630-f006:**
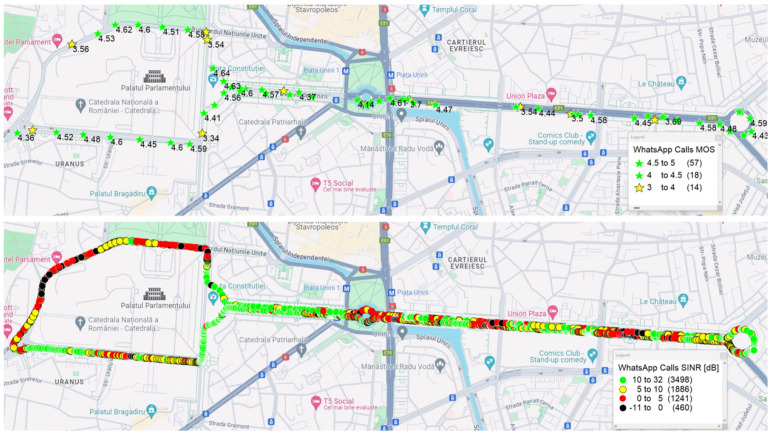
WhatsApp drive test: correlation between MOS (**top**) and SINR (**bottom**).

**Figure 7 sensors-24-06630-f007:**
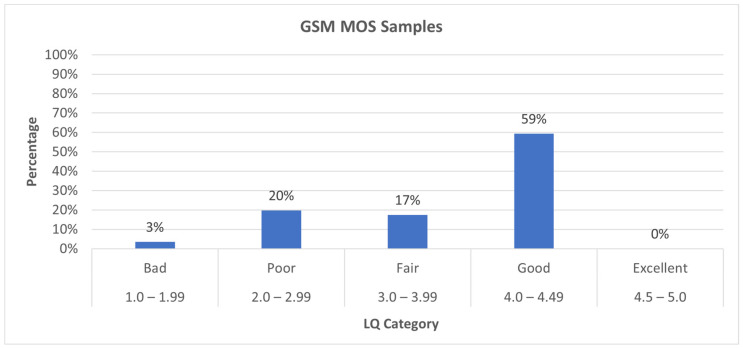
GSM MOS value percentage distribution.

**Figure 8 sensors-24-06630-f008:**
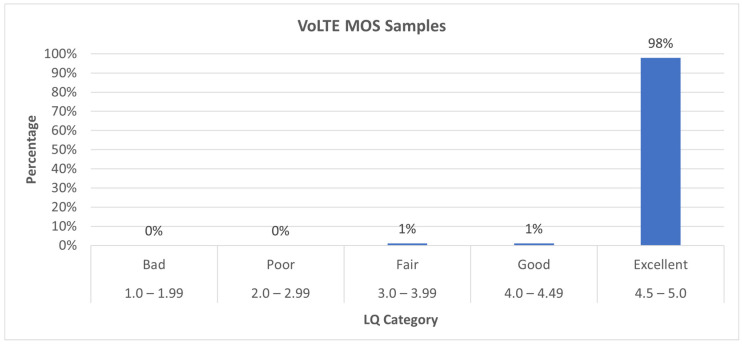
VoLTE MOS value percentage distribution.

**Figure 9 sensors-24-06630-f009:**
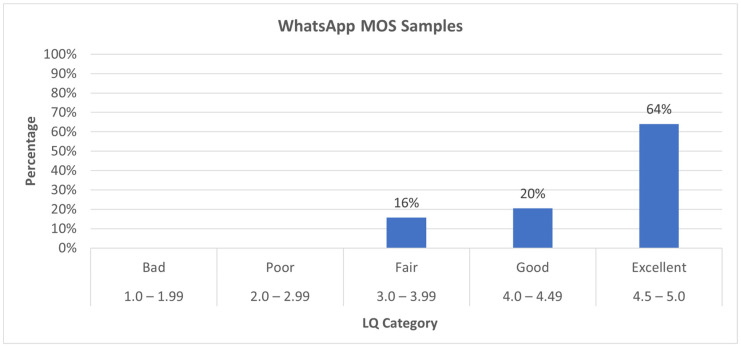
WhatsApp MOS value percentage distribution.

**Table 1 sensors-24-06630-t001:** Correspondence MOS values and LQ categories.

MOS Values	Quality Category
1.0–1.99	Bad
2.0–2.99	Poor
3.0–3.99	Fair
4.0–4.49	Good
4.5–5.0	Excellent

**Table 2 sensors-24-06630-t002:** Call window configuration.

Parameter	Value [s]
Window duration	130
Call duration	80
Pause duration	20
1st call setup delay	5
Guard time	5
Max call setup time	30
Call alerting timeout	0
Call answer timeout	0
Call pickup delay	3.0
No call setup time	115
Max failed calls	2
Pause after failed call	30
No connection timeout	600

**Table 3 sensors-24-06630-t003:** Categories of MOS results.

Call Scenario	No. Samples Bad	No. Samples Poor	No. Samples Fair	No. Samples Good	No. Samples Excellent
GSM	3	17	15	51	
VoLTE			1	1	93
WhatsApp			14	18	57

**Table 4 sensors-24-06630-t004:** Quality degradation causes for every type of MOS categories in the GSM scenario.

Quality Category	Degradation Cause					
	Background Noise	Frame/Packet Loss	Interruptions	Variable Delay	Not to Specify	OK
Poor		1	13	1	2	
Bad		1	2			
Fair	1		1		4	9
Good						51

**Table 5 sensors-24-06630-t005:** Quality degradation causes for every type of MOS category in the WhatsApp scenario.

Quality Category		Degradation Cause	
	Frame/Packet Loss	Variable Delay	OK
Fair	9	3	2
Good			17
Excellent			57

## Data Availability

Data are contained within the article.

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
