# Peer review of "Voice Quality Evaluation in a Mobile Cellular Network: In Situ Mean Opinion Score Measurements"

_sensors, 2024, doi:10.3390/s24206630_

Round 1
Reviewer 1 Report
Comments and Suggestions for Authors
The paper presents an experimental measurement of voice quality parameters in 2G and 4G networks in Bucharest. The topic is interesting, although it could attract much more attention for general readers if the data quality rate and other parameters associated to the data transmission would be considered.
The manuscript focus only on some measurements and its length also does not seem appropriate for a research publication in this journal.
I also would suggest to focus in 4G and also in the new 5G networks in the country, as 2G is basically being dismissed around the world.
The experiments as presented hardly could be replicated in other cities of the same country or even in other countries, which compromise the possibility of replication of your work for other researchers.
Basically there is no theoretical or state of the art review, as the manuscript is focused only in voice quality.
The results and data presented are not relevant as it is presented. And there is no deep conclusions based on the test that can be extended to other mobile networks. Figures 7 to 9 do not aggregate anything new or relevant to the current status of the mobile networks - neither to academia studies nor to industry applications.
The references presented must be completely updated, as basically there aren't relevant references cited by the authors.
Comments on the Quality of English LanguageThere are some typos that should be revised in the manuscript, although the overall quality of English language is appropriate.
Author Response
Comments 1: The paper presents an experimental measurement of voice quality parameters in 2G and 4G networks in Bucharest. The topic is interesting, although it could attract much more attention for general readers if the data quality rate and other parameters associated to the data transmission would be considered.
Response 1: Thank you for your helpful feedback. In section Introduction (lines 79-90) and Final Remarks (lines 496-551), we have emphasized that the results are of interest not only to the general public but also to radio engineers who optimize real radio access networks. Additionally, we expanded the analysis of the results by including causes for lower MOS scores (e.g., bad, poor, fair), which are now detailed in the Results section (lines 371-376, 382-421, 429-439).
Comments 2: The manuscript focus only on some measurements and its length also does not seem appropriate for a research publication in this journal.
Response 2: In the revised version of the manuscript, we have made several key additions to address your concerns. First, in section Key Strength of Our Work (lines 169-192) (lines 21-23) we indicate the originality and the key strength of our work. We emphasize that the study is based on real-world, in-situ measurements, providing practical data that reflect actual network performance under varying conditions. Additionally, we used professional measurement system developed by Rohde & Schwarz (lines 310-315), which is widely employed by mobile network operators, ensuring the reliability and relevance of our results. We also expanded the Results lines 371-475, 382-421, 429-439) and Conclusion (lines 441-511) sections to provide a more comprehensive discussion of the findings. This includes a broader comparison of three distinct voice technologies — GSM, VoLTE, and OTT (WhatsApp) — which adds depth to the analysis and demonstrates the broader applicability of the study for both academic research and industry practices.
Comments 3: I also would suggest to focus in 4G and also in the new 5G networks in the country, as 2G is basically being dismissed around the world.
Response 3: In Europe, and implicitly in Romania, most operators have shut down their 3G networks, opting to retain GSM networks due to their lower spectrum requirements and better coverage (lines 474-484, 492-495). The penetration of VoLTE services in Romania and across European countries is approximately 75% of all voice service users. As a result, a significant portion of users, when connected to 4G and attempting to make a call, are transferred to the GSM network through the CSFB procedure in the CS domain. GSM continues to play an important role for both users and operators. For Voice over New-Radio (Vo-NR), in Romania and most European countries, mobile network operators have not yet implemented 5G Standalone technology, as the required investments in the Core network are substantial. As a result, this service is expected to remain unavailable for the next 2 to 3 years. We also provided an analysis of 5G, describing its current limitations in Romania’s NSA configuration and why traffic was routed through 4G. This explanation is included (lines 303-309).
Comments 4: The experiments as presented hardly could be replicated in other cities of the same country or even in other countries, which compromise the possibility of replication of your work for other researchers.
Response 4: The measurement system used is professional and widely employed by most mobile network operators. The system's documentation is publicly available and can be downloaded online. The drive tests were conducted in a car, with the phones positioned on the dashboard to simulate typical user behavior. The call sequence configuration is detailed in Table 2 (line 288). These drive tests can be replicated at any time in other cities or by other research teams. In fact, it is recommended to conduct drive tests in different locations and compare the measurements.
Comments 5: Basically, there is no theoretical or state of the art review, as the manuscript is focused only in voice quality.
Response 5: We have updated the State of Art (lines 96-168) section by incorporating additional studies. We discussed the limitations and gaps in those works, which our study addresses. Furthermore, we highlighted the novel contributions of our work in contrast to the existing literature. We add the section Key strengths of our work (lines 169-192).
Comments 6: The results and data presented are not relevant as it is presented. And there is no deep conclusions based on the test that can be extended to other mobile networks. Figures 7 to 9 do not aggregate anything new or relevant to the current status of the mobile networks - neither to academia studies nor to industry applications.
Response 6: In the revised version of the manuscript, we have expanded both the MOS Performance Evaluation (lines 371-376, 382-421, 429-439) and Conclusions (lines 441-511) sections to provide a more thorough analysis of the data and findings. We have included additional details that better explain the results and offer deeper insights into how these findings can be applied to other mobile networks. By providing more context and elaborating on the implications of the test results, we believe the conclusions now offer a stronger basis for extending the analysis to other network environments.
Comments 7: The references presented must be completely updated, as basically there aren't relevant references cited by the authors.
Response 7: The references have been updated to include more recent and relevant studies. These additions are reflected throughout the manuscript and in the Reference (line 547-549, 556-575, 579-583) list ensuring a comprehensive citation of state-of-the-art research in the field.

Reviewer 2 Report
Comments and Suggestions for Authors
Please see the attachment.

Author Response
General Comments : In this article, the authors wrote a topic titled ‘Voice Quality Evaluation in a Mobile Cellular Network: In-situ 2 MOS Measurements’ This topic is current and the area of research is relevant. The authors did their measurement report considering GSM to LTE technology. The performance of these past technologies, though still in use, are already well-research with enough literature on them for the past tens years. Thus, I suggest that the authors also adding measurement from 5G new radio technology, which is the relatively new and being the current technology. I think the readers will be more interested in it.
Response: In Europe, and implicitly in Romania, most operators have shut down their 3G networks, opting to retain GSM networks due to their lower spectrum requirements and better coverage (lines 474-484, 492-495). The penetration of VoLTE services in Romania and across European countries is approximately 75% of all voice service users. As a result, a significant portion of users, when connected to 4G and attempting to make a call, are transferred to the GSM network through the CSFB procedure in the CS domain. GSM continues to play an important role for both users and operators. For Voice over New-Radio (Vo-NR), in Romania and most European countries, mobile network operators have not yet implemented 5G Standalone technology, as the required investments in the Core network are substantial. As a result, this service is expected to remain unavailable for the next 2 to 3 years. We also provided an analysis of 5G, describing its current limitations in Romania’s NSA configuration and why traffic was routed through 4G. This explanation is included (lines 303-309).
Comments 1: Abstract - I suggest that the authors should add the importance of their work to the abstract section as revealed in line 76-78.
Response 1: We followed your suggestion, and we updated the Abstract section (lines 21-23). Also, in the revised version of the manuscript, we have expanded both the MOS Performance Evaluation (lines 371-376, 382-421, 429-439) and Conclusions (lines 441-511) sections to provide a more thorough analysis of the data and findings. We have included details that better explain the results and offer deeper insights into how these findings can be applied to other mobile networks. By providing more context and elaborating on the implications of the test results, we believe the conclusions now offer a stronger basis for extending the analysis to other network environments. Additionally, in section Key Strength of Our Work (lines 169-192) (lines 21-23) we indicate the originality and the key strength of our work.
Comments 2: Introduction - This section is somewhat satisfactory, but I strongly suggest that the authors should provide clear and strong research gap in this section with outlined contribution to knowledge.
Response 2: We have added a clearer statement of the research gap and outlined the contributions of this study in the Introduction section (lines 79-88). This includes how our work addresses gaps in current literature, particularly in real-world in-situ testing and network performance evaluation.
Sections 2 and 3 - Provide a reference for table 1. Figure 2 is not clear should be improved or replaced. Detailed information of the equipment used for measured should be provided. Information on about the transmission frequency/bandwidth, transmitters, measurement locations should be provided in tabular form
Response 3: We have revised Figure 2 (line 280) to improve clarity and provided the requested references for Table 1 (line 269). Also, we have clearly specified the instrumentation and measurement setup (lines 272-276). We used a professional measurement system widely employed by mobile operators, developed by Rohde & Schwarz, to ensure the accuracy and reliability of the data. As described, the drive tests were performed using two smartphones running on Android operating systems, positioned on the dashboard of the vehicle to simulate real user behavior during typical mobile network usage. Additionally, we provided the file used for the MOS evaluation, named “En_P501_FM_wide_ref.waw”, and gave detailed instructions on the call window in Table 2 (line 287) used during testing. Furthermore, we conducted “Double Ended Call” test, a method specific to the measurement system, with two smartphones to ensure accuracy and consistency, using the same call sequence across all scenarios (2G, VoLTE, and WhatsApp). This approach guarantees that the results are robust and can be replicated in other environments using the same methodology and equipment.
Sections 4 and 5 - Provide detail statistics and explanation on acquired data sample
Response 4: In response to your suggestion, we have added a detailed explanations in the MOS Performance Evaluation (lines 371-376, 382-421, 429-439) section regarding the statistics and processing of the acquired data samples. Specifically, for the three scenarios — 2G, 4G VoLTE, and OTT WhatsApp — we clarified that the logs were downloaded using the proprietary Rohde & Schwarz Smart Analytics system. After extraction, the data was processed into graphs and maps, as presented in the paper, providing a clear visual representation of the results.
Conclusion - The summarized results of the entire work should be provided in this section. The contribution to knowledge should also be revealed. Future works in connection to this presented one should be revealed.
Response 5: In the revised version of the manuscript, we have completely rewritten the Conclusions (lines 441-511) and Final Remarks (lines 496-511) sections to provide a more in-depth analysis of the results. These offers a clearer summary of the key findings and their implications for network performance.

Reviewer 3 Report
Comments and Suggestions for Authors
The authors in this paper evaluate the quality of voice service through a drive test for 2G and 4G cellular mobile networks. Voice quality assessments composes three scenarios, which are: over the CS domain using 2G technology, over the PS domain using VoLTE and OTT WhatsApp application in 4G technology. The paper in general is well written and organized, but unfortunately lacks to the following basic points:
- The paper doesn’t provide any critique to literature and didn’t address properly where/how related work performed voice quality evaluations and what were their assessments outcomes.
- The paper lacks to a comparison section that highlights the differences between the collected results and other related evaluation procedures that presented in literature. It may not be necessary that the assessments have been carried in the same city/country, but require the comparison only to show if the compared technologies performed similarly in different sites.
- The paper didn’t justify the obtained results and didn’t manage to discuss the presented outcomes clearly.
- The conclusion section has indicated some general information as “WhatsApp Calls are convenient, versatile, and cost-effective for internet-based communication”, “GSM calls can have satisfactory quality and can ensure call continuity from 4G through the SRVCC procedure” and “Both VoLTE and WhatsApp services offer superior speech quality”. It seems that the authors in the conclusion section has only mentioned some a de-facto facts regarding the voice quality, and didn’t manage to provide any original contributions related to their findings from their evaluation.
Comments on the Quality of English Language
Some minor English language editing is required
Author Response
Comments 1: The authors in this paper evaluate the quality of voice service through a drive test for 2G and 4G cellular mobile networks. Voice quality assessments composes three scenarios, which are: over the CS domain using 2G technology, over the PS domain using VoLTE and OTT WhatsApp application in 4G technology. The paper in general is well written and organized, but unfortunately lacks to the following basic points:
Response 1: We thank the reviewer for taking the time to analyze our work and provide us with valuable advice on how to improve our research. Please find our response below.
Comments 2: The paper doesn’t provide any critique to literature and didn’t address properly where/how related work performed voice quality evaluations and what were their assessments outcomes.
Response 2: We have updated the State of Art (lines 96-168) section by incorporating additional studies. We discussed the limitations and gaps in those works, which our study addresses. Furthermore, we highlighted the novel contributions of our work in contrast to the existing literature. We add the section Key strengths of our work (lines 169-192).
Comments 3: The paper lacks to a comparison section that highlights the differences between the collected results and other related evaluation procedures that presented in literature. It may not be necessary that the assessments have been carried in the same city/country, but require the comparison only to show if the compared technologies performed similarly in different sites.
Response 3: In the revised manuscript, we have emphasized that all measurements were conducted according to a standardized methodology (lines 272-276, 281-285), and the collected data was processed in compliance with these standards. This ensures the transparency and replicability of the methodology. Furthermore, as noted in the Experimental Setup section (lines 308-313), we used a professional measurement system developed by Rohde & Schwarz, which is widely employed by mobile network operators to ensure the accuracy and reliability of the data. The drive tests were carried out using two smartphones running on Android operating systems, positioned on the dashboard of the vehicle to simulate real user behavior during mobile network usage. Additionally, the file used for the MOS evaluation, named “En_P501_FM_wide_ref.wav”, was specified, along with detailed instructions on the call window used during testing, as shown in Table 2 (line 287). We also conducted the “Double Ended Call” test (line 286), a method specific to the measurement system, with two smartphones to ensure accuracy and consistency across all scenarios (2G, VoLTE, and WhatsApp). This robust approach guarantees that the results can be replicated in other environments using the same methodology and equipment. It is also important to note that these measurements must be conducted continuously, as network conditions and testing environments frequently change. This is a common and ongoing process for mobile network operators, aimed at optimizing and configuring the network to consistently deliver high-quality service to users. Thus, the focus is not solely on comparing results with other studies, but rather on how the network responds to changes and how it can be optimized to maintain satisfactory service levels.
Comments 4: The paper didn’t justify the obtained results and didn’t manage to discuss the presented outcomes clearly.
Response 4: In the revised version of the manuscript, we have expanded both the MOS Performance Evaluation (lines 371-376, 382-421, 429-439) and Conclusions (lines 441-511) sections to provide a more thorough analysis of the data and findings. We have included additional details that better explain the results and offer deeper insights into how these findings can be applied to other mobile networks. By providing more context and elaborating on the implications of the test results, we believe the conclusions now offer a stronger basis for extending the analysis to other network environments.
Comments 5: The conclusion section has indicated some general information as “WhatsApp Calls are convenient, versatile, and cost-effective for internet-based communication”, “GSM calls can have satisfactory quality and can ensure call continuity from 4G through the SRVCC procedure” and “Both VoLTE and WhatsApp services offer superior speech quality”. It seems that the authors in the conclusion section has only mentioned some a de-facto facts regarding the voice quality, and didn’t manage to provide any original contributions related to their findings from their evaluation.
Response 5: In the revised version of the manuscript, we have completely rewritten the Conclusions (lines 441-511) and Final Remarks (lines 496-511) sections to provide a more in-depth analysis of the results. These offers a clearer summary of the key findings and their implications for network performance.

Round 2
Reviewer 1 Report
Comments and Suggestions for Authors
I congratulate the authors as the revised version of the manuscript was significantly improved considering the original version. Most part of the problems identified in the original version were fixed by the authors and also apropriately adressed in the cover letter.
Some suggestions to improve this revised version:
- please adjust the length of the Figures 1, 3, 5 and 6.
- please improve the quality of the Figure 2. As it is presented, it is hard to read what is presented in the screens provided.
- in my opinion, the conclusions now are very extense. Maybe the subsections 7.1, 7.2, 7.3 and 7.4 could be merged to the section 6.
Author Response
Comment 1: I congratulate the authors as the revised version of the manuscript was significantly improved considering the original version. Most part of the problems identified in the original version were fixed by the authors and also apropriately adressed in the cover letter.
Response 1:
We thank the reviewer for taking the time to carefully review the revised paper and for acknowledging the improvements.
Comment 2: Some suggestions to improve this revised version:
- please adjust the length of the Figures 1, 3, 5 and 6.
- please improve the quality of the Figure 2. As it is presented, it is hard to read what is presented in the screens provided.
- in my opinion, the conclusions now are very extense. Maybe the subsections 7.1, 7.2, 7.3 and 7.4 could be merged to the section 6.
Response 2:
We thank the reviewer for the valuable feedback. The paper has been revised and the following changes were made:
- Figures 1, 3, 5, and 6 were resized to match the width of the text column.
- Figure 2 was replaced. The new version has an enhanced quality to ease readability and, besides the part that shows the configuration window, it includes a second part to illustrate the call window during operation of the two DUTs. An explanatory text was added (lines 279-280) and the figure caption (line 282) was updated accordingly.
- In this revised version of the manuscript, Section 6 is now divided in 2 subsections: “6.1. MOS Evaluation” that begins at line 317 and “6.2 Performance evaluation” which begins at line 441. Paragraphs 7.1, 7.2, 7.3, and 7.4 were moved to subsection 6.2 (lines 442 – 487). Now the Conclusions section has a more appropriate length, synthesizing just the key findings of the paper (lines 489 – 508).

Reviewer 2 Report
Comments and Suggestions for Authors
The manuscript is now in a better form as the authors have tackled the various issues I raised during the first review
Author Response
General Comment: The manuscript is now in a better form as the authors have tackled the various issues I raised during the first review.
Response:
We thank the reviewer for taking the time to carefully review the revised paper and for acknowledging the improvements.

Reviewer 3 Report
Comments and Suggestions for Authors
Based on the amendments that have been carried to the revised manuscript, its clear that the authors have managed to properly address all highlighted comments.
Author Response
General comment: Based on the amendments that have been carried to the revised manuscript, its clear that the authors have managed to properly address all highlighted comments:
Response: We thank the reviewer for taking the time to carefully review the revised paper and for acknowledging the improvements.
